

# Development of rapid and cost-effective multiplex PCR assays to differentiate catfish of the genus *Brachyplatystoma* (Pimelodidae–Siluriformes) sold in Brazil

Leilane Freitas[1], Andressa J. Barbosa[1], Bianca A. Vale[1], Iracilda Sampaio[2] and Simoni Santos[1]

[1] Laboratory of Fish Microbiology, Institute of Coastal Studies, Universidade Federal do Pará, Bragança, PA, Brasil
[2] Laboratory of Evolution, Institute of Coastal Studies, Universidade Federal do Pará, Bragança, PA, Brasil

## ABSTRACT

The catfishes *Brachyplatystoma filamentosum* (Kumakuma), *Brachyplatystoma vaillantii* (Laulao catfish), and *Brachyplatystoma rousseauxii* (gilded catfish) are important fishery resources in Brazil, where they are sold both fresh and in the form of fillets or steaks. These species have morphological similarities, thus, they can be easily misidentified or substituted, especially after processed. Therefore, accurate, sensitive, and reliable methods are needed for the identification of these species to avoid commercial fraud. In the present study, we develop two multiplex PCR assays for the identification of the three catfish species. Each multiplex protocol combined three species-specific forward primers and a universal reverse primer to produce banding patterns able to discriminate the target species unequivocally. The length of the cytochrome C oxidase subunit I (COI) fragments was approximately 254 bp for *B. rousseauxii*, 405 bp for *B. vaillantii*, and 466 bp for *B. filamentosum*, while the control region (CR) assay produced fragments of approximately 290 bp for *B. filamentosum*, 451 bp for *B. vaillantii*, and 580 bp for *B. rousseauxii*. The protocols were sensitive enough to detect the target species at a DNA concentration of 1 ng/µL, with the exception of the CR of *B. vaillantii*, in which the fragment was only detectable at 10 ng/µL. Therefore, the multiplex assays developed in the present study were sensitive, accurate, efficient, rapid, and cost-effective for the unequivocal identification of the target species of *Brachyplatystoma*. They can be utilized by fish processing industries to certify their products, or by government agencies to authenticate products and prevent fraudulent commercial substitutions.

Corresponding author
Simoni Santos, simoni@ufpa.br

## INTRODUCTION

The catfish of the genus *Brachyplatystoma* are members of the family Pimelodidae (Siluriformes), which is endemic to the rivers of the Neotropical region, principally the Amazon and Orinoco basins (*Lundberg, Sullivan & Hardman, 2011*; *Fricke, Eschmeyer & Van der Laan, 2022*). The genus has seven species—*Brachyplatystoma rousseauxii* Castelnau, 1855, *Brachyplatystoma vaillantii* Valenciennes, 1840, *Brachyplatystoma filamentosum* Lichtenstein, 1819, *Brachyplatystoma juruense* Boulenger, 1898, *Brachyplatystoma platyne­mum* Boulenger, 1898, *Brachyplatystoma tigrinum* Britski, 1981, and *Brachyplatystoma capapretum* Lundberg & Akama, 2005—of which, the former three are the most important commercial fishery resources in Brazil (*IBAMA, 2005–2007*; *MPA, 2012a*; *MPA, 2012b*).

While no recent fisheries statistics are available for Brazil, the official records indicate that *B. vaillantii* (Laulao catfish, piramutaba in Brazil), *B. rousseauxii* (gilded catfish, dourada in Brazil), and *B. filamentosum* (Kumakuma, filhote in Brazil) are among the principal freshwater species exploited by Brazilian commercial fisheries, with a total catch of 295,221.2 tons (t) being landed between 2005 and 2011 (*IBAMA, 2005–2007*; *MPA, 2012a*; *MPA, 2012b*). The bulk of this catch refers to *B. vaillantti* (171,939 t), followed by *B. rousseauxii* (100,326 t), and *B. filamentosum*, with 22,956.2 t (*IBAMA, 2005–2007*; *MPA, 2012a*; *MPA, 2012b*). Despite its smaller catches, *in loco* observations indicate that *B. filamentosum* is the most highly valued of the three species, especially when sold as frozen fillets or steaks, with a mean retail price of BRL 72.13/kg, followed by *B. rousseauxii* (BRL 41.20/kg) and *B. vaillantii* (BRL 35.08/kg).

When fresh, these three species can be distinguished based on their coloration and the shape and configuration of the body, adipose fin, and maxillary barbels (*Lundberg & Littmann, 2003*; *Santos, Ferreira & Zuanon, 2009*), although this is not an easy task for non-experts, which may often lead to their misidentification when marketed for human consumption. This problem is accentuated when the fish is sold in the form of frozen fillet or steaks, which removes the morphological characteristics necessary for the visual discrimination of the species. This favors substitutions and mislabeling during the processing of the fish products (*But, Wu & Shaw, 2019*; *Carvalho, Sampaio & Santos, 2020*).

Previous studies have evaluated the authenticity of processed *Brachyplatystoma* fillets using DNA barcoding and recorded a global substitution rate of 18% (*Carvalho, 2018*; *Carvalho, Sampaio & Santos, 2020*). This rate increased to 29% in the case of the filhote (*B. filamentosum*), which is the most expensive of the three species, followed by the dourada (26%) and lastly, the piramutaba (9%), the least expensive species. The results of the study indicate that the vast majority of the mislabeled fish were less expensive than the species indicated on their labels, which is clear evidence of commercial fraud (*Carvalho, 2018*; *Carvalho, Sampaio & Santos, 2020*).

The substitution of species may have economic impacts and also affect human health, when the substitute species contain allergens or toxins (*Giusti et al., 2016*; *Roungchun, Tabb & Hellberg, 2022*). Substitutions may also have ecological impacts, when the substitute taxa are endangered, overfished or have depleted stocks (*Brito et al., 2015*; *Barbosa et al., 2020*;

*Carvalho, Sampaio & Santos, 2020*; *Barbosa, Sampaio & Santos, 2021*). Clearly, any method that can guarantee the systematic identification and authentication of species will be an extremely valuable tool for the prevention of fraud and the avoidance of the associated impacts.

In recent years, DNA-based methods have been applied increasingly to the effective discrimination of taxa, even in the case of highly-processed products. These methods range from more conventional procedures, such as PCR followed by Restriction Fragment Length Polymorphism (PCR-RFLP), DNA sequencing, and multiplex PCR (*Abbas et al., 2018*; *Lo & Shaw, 2018*; *Böhme et al., 2019*), to emerging new molecular techniques, such as real-time PCR (*Kang, 2019*; *Brenn et al., 2021*), DNA Barcode High Resolution Melting (BAR-HRM), droplet digital PCR (ddPCR), isothermal amplification, DNA array, and next-generation sequencing, or NGS (*Abbas et al., 2018*; *Fernandes et al., 2018*; *Lo & Shaw, 2018*; *Böhme et al., 2019*; *Franco et al., 2021*; *Noh et al., 2021*).

While all these methods are extremely efficient for the identification of species and the authentication of fish products, the effective identification of frauds in the food sector, particularly in developing countries, requires rapid, efficient, sensitive, and cost-effective techniques (*Sangthong, Ngernsiri & Sangthong, 2014*; *Castigliego et al., 2015*; *Giusti et al., 2016*; *Veneza et al., 2017*; *Barbosa et al., 2020*; *Barbosa, Sampaio & Santos, 2021*). One potential solution here is multiplex PCR assay, which does not require any major laboratory infrastructure, and involves only a few procedural steps. This technique involves the combination of different pairs of species-specific primers, which permit the unequivocal discrimination of the target taxa in the same reaction through the amplification of distinct banding patterns, which can be visualized in agarose gel (*Ali et al., 2012*; *Böhme et al., 2019*). In recent years, several studies have developed multiplex PCR protocols for species identification, and have confirmed their accuracy, sensitivity, and robustness for the identification and authentication of fish samples (*Veneza et al., 2017*; *Kang, 2019*; *Barbosa et al., 2020*; *Barbosa, Sampaio & Santos, 2021*; *Wilwet et al., 2021*). Among the markers used for the development of multiplex protocols for species identification purposes, mitochondrial DNA markers are known for their advantages such as the large number of copies per sample, the availability of sequences in public databases, as well as genomic regions with intraspecific conservation and interspecific polymorphism, which are necessary characteristics for the development of species-specific primers (*Hebert, Ratnasingham & DeWaard, 2003*; *Kang, 2019*; *Böhme et al., 2019*).

Overall, then, considering the economic importance of the catfish of the genus *Brachyplatystoma* (*B. vaillantii*, *B. rousseauxii*, and *B. filamentosum*) and the known occurrence of fraudulent mislabeling by retailers in Brazil (*Carvalho, 2018*; *Carvalho, Sampaio & Santos, 2020*), it is fundamentally important to develop a rapid, accurate, reliable, and cost-effective methods that can guarantee the identification of fish products, and mitigate the potential impacts of these substitutions. In the present study, two multiplex PCR assays were developed for the discrimination of *B. vaillantii*, *B. rousseauxii*, and *B. filamentosum*, which can be used for the identification and authentication of products derived from these catfish.

## MATERIALS & METHODS

### Sampling

A total of 31 samples of muscle tissue were extracted from whole fish obtained from local fish markets in the Brazilian state of Pará, including 11 samples of *B. filamentosum*, 13 of *B. rousseauxii*, and seven of *B. vaillantii* (Table 1). These samples were used for the development and validation of the multiplex protocols described in the present study. Samples of muscle tissue were also obtained from 11 specimens of pimelodid taxa that are commonly sold together with the target species in local fish markets, which were used to test the specificity of the primers (Table 1). These taxa were *Brachyplatystoma platynemum* ($n = 2$ specimens), *Phractocephalus hemioliopterus* Bloch & Schneider, 1801 ($n = 3$), *Zungaro zungaro* Humboldt, 1821 ($n = 3$), *Pseudoplatystoma fasciatum* Linnaeus, 1766 ($n = 1$), and *Pseudoplatystoma reticulatum* Eigenmann & Eigenmann, 1889 ($n = 2$). In addition, for the primer design a further 36 COI and 32 Control Region (CR) sequences were downloaded from GenBank, including the target species, their congeners and other pimelodid taxa (Table 1). The samples of muscle tissue were preserved in absolute ethanol and stored in the tissue bank of the Laboratory of Fishery Microbiology at the UFPA Coastal Studies Institute in Bragança.

### Isolation, amplification, sequencing of the DNA and the identification of the samples

The total DNA was extracted using the Wizard Genomic DNA Purification kit (Promega, Madison, WI, USA), following the protocol of the manufacturer for muscle tissue. The concentration and purity of the DNA were evaluated in a Nanodrop 2000 spectrophotometer (Thermo Scientific, Waltham, MA, USA).

The barcoding region of the COI gene was amplified by Polymerase Chain Reaction (PCR) with FishF1 and FishR1 primers (*Ward et al., 2005*), using the protocol and cycling program previously described in *Carvalho, Sampaio & Santos (2020)*.

The positive reactions were purified using the polyethylene glycol (PEG 8000) protocol (*Paithankar & Prasad, 1991*) and sequenced by the dideoxyterminal method of *Sanger, Nicklen & Coulson (1977)* using the BigDye Terminator v3.1 Cycle Sequencing kit (Applied Biosystems, Austin, TX, USA), with the electrophoresis being run in an ABI 3500 XL (Applied Biosystems, Austin, TX, USA).

The sequences were aligned and edited in Bioedit 7.0.9.0 (*Hall, 1999*), and the samples of *B. filamentosum, B. rousseauxii*, and *B. vaillantii* were identified based on the reference sequences published by *Carvalho, Sampaio & Santos (2020)*. The sequences of the other pimelodid species were identified by comparison with the sequences available in the public databases, GenBank (using BLAST: https://blast.ncbi.nlm.nih.gov/Blast.cgi) and the BOLD System (https://www.boldsystems.org/). In all cases, individuals were considered to be members of the same species when their genetic similarity was ≥98%, the criterion adopted by the BOLD System for the inclusion of reference specimens in the Barcode Database Reference (*Ward, Hanner & Hebert, 2009*).

**Table 1** Sequences of fish species of the family Pimelodidae used to design the species-specific primers.

| Species | *n* (COI) | Accession number for COI | *n* (CR) | Accession number for the Control Region |
|---|---|---|---|---|
| *Brachyplatystoma rousseauxii* | 15 | MT551748; FJ418759; OQ576109–OQ576121 | 3 | DQ779044–DQ779046 |
| *Brachyplatystoma vaillantii* | 11 | MT551751–MT551753; KT952409; OQ576122–OQ576128 | 3 | HQ444957–HQ444959 |
| *Brachyplatystoma filamentosum* | 14 | MT551754–MT551756; OQ576098–OQ576108 | 3 | GU442097; GU442117; GU442118 |
| *Brachyplatystoma tigrinum* | 1 | KT952408 | – | – |
| *Brachyplatystoma juruense* | 2 | KT952405; KR491564 | – | – |
| *Brachyplatystoma platynemum* | 3 | KT952406; OQ576135; OQ576136 | 3 | MK779967–MK779969 |
| *Brachyplatystoma capapretum* | 1 | KT952403 | 3 | GU903438; GU903439; GU903460 |
| *Pseudoplatystoma reticulatum* | 4 | GU570868; GU570869; OQ576137; OQ576138 | 4 | FJ024076–FJ024078; KU291530 |
| *Pseudoplatystoma magdaleniatum* | 2 | GU570859; GU570860 | 2 | NC026526; KP090204 |
| *Pseudoplatystoma fasciatum* | 2 | GU570849; OQ576139 | – | – |
| *Pseudoplatystoma corruscans* | 3 | HQ600841; HQ600842; JX462914 | 5 | FJ024079; FJ024070; FJ024071; NC026846; KJ502112 |
| *Pseudoplatystoma metaense* | 1 | JQ733555 | – | – |
| *Pseudoplatystoma orinocoense* | 1 | JQ733556 | – | – |
| *Pseudoplatystoma tigrinum* | 2 | GU570936; GU570937 | – | – |
| *Pseudoplatystoma punctifer* | 1 | KT952427 | – | – |
| *Sorubimichthys planiceps* | 2 | GU570940; GU570941 | – | – |
| *Hemisorubim platyrhynchos* | 2 | GU570707; KT952411 | – | – |
| *Zungaro zungaro* | 5 | KT952431; KP294234; OQ576129–OQ576131 | 3 | FJ797695–FJ797697 |
| *Zungaro jahu* | 2 | EU179810; JN813033 | 2 | GQ254528, GQ254529 |
| *Phractocephalus hemioliopterus* | 4 | KP772589; OQ576132–OQ576134 | – | – |
| *Sorubim cuspicaudus* | – | – | 1 | NC026211 |

**Notes.**

*n*, sample size; –, no data.

## Design of the species-specific primers

Two databases were used to design the species-specific primers. The COI and CR databases were selected after evaluation the sequences of different genomic regions available in GenBank since they presented the number of samples of the target taxa and closely related pimelodid, making it possible to evaluate the variability of the markers and find regions of intraspecific conservation and interspecific variability, in the target species, for the design of the primers. The COI alignment had 592 base pairs (bp) and contained at least five sequences from each target species, generated in the present study, and 36 sequences obtained from GenBank (Table 1). The mitochondrial CR alignment consisted of 32 sequences of 950 bps, including at least three sequences of each target *Brachyplatystoma* species and other pimelodids obtained from GenBank (Table 1). For the two markers, we evaluated a larger number of sequences from the target species and the closely related Pimelodidae taxa, whenever available, and carefully selected sequences from different

haplotypes to expand the possibility of capturing possible intraspecific variations and thus obtaining more specific primers.

Following the automatic alignment in BioEdit, the sequences were inspected visually to identify species-specific sites, for which the forward primers were designed. When combined with a universal reverse primer, these forward primers generate fragments of a distinct size for each species in the multiplex PCR assay. The reverse primers were FishR1 for the COI (*Ward et al., 2005*; Table 2) and F12R for the CR (*Sivasundar, Bermingham & Ortí, 2001*; Table 3). A pair of primers of the 16S Ribosomal DNA (rDNA) gene were also selected to amplify a control band in each multiplex protocol (Tables 2 and 3).

The Multiple Primer Analyzer program (Thermo Fisher Scientific) was used to evaluate the melting temperature (Tm), auto-complementarity, GC content of the 3′ portion, and the dimers of the set of primers used for each protocol. The specific primers that presented the best parameters in this analysis were selected for synthesis by the Síntese Biotecnologia company of Belo Horizonte, Brazil (Tables 2 and 3).

## Development of the multiplex PCR assays

The specificity of the primers was tested using a simplex PCR, in order to eliminate the possibility of unspecific amplification. This test was based on reactions using five specimens of each target species, together with at least one specimen of each of the five pimelodid species that are most often sold together with the target species, *i.e., B. platynemum, P. hemioliopterus, Z. zungaro, P. fasciatum*, and *P. reticulatum*. The PCR reactions for the specificity test were run in a final volume of 15 $\mu$L containing 2 $\mu$L of dNTPs (1.25 mM; Sinapse Biotecnologia, São Paulo, SP, Brazil), 1.6 $\mu$L of buffer solution (10x) containing $MgCl_2$ (25 mM), 0.35 $\mu$L of $MgCl_2$ (25mM), 0.6 $\mu$L of BSA (5 mg/mL; Thermo Fisher Scientific, Vilnius, Lithuania), 0.15 $\mu$L of each primer (10 pmol/$\mu$L), 1.0 $\mu$L of the total genomic DNA (100 ng/$\mu$L), 0.2 $\mu$L of JumpStart Taq DNA polymerase (2.5 U/$\mu$L) (Sigma-Aldrich, St. Louis, MO, USA), and pure water to complete the final volume. In these tests, the species-specific forward primers were combined with the reverse primers, that is, FishR1 for the COI gene (*Ward et al., 2005*) and F12R for the mitochondrial Control Region (*Sivasundar, Bermingham & Ortí, 2001*), which are common to all the species (Tables 2 and 3). A negative control, containing pure water, was included in all the reactions. The fragments were amplified in a gradient Thermal Cycler TX96 Plus (Amplitherm, Cotia, Brazil).

The simplex PCRs were run for both markers with an initial denaturation at 94 °C for 4 min, followed by 35 cycles of denaturation at 94 °C for 30 s, hybridization at 60.5 °C for 30 s, and extension at 72 °C for 1 min, with a final extension at 72 °C for 7 min. From the five samples tested, three samples of each marker from each target species were then purified and sequenced to confirm the identification of the species.

The multiplex PCR assays for each marker included all the species-specific primers and the universal reverse primer. In this step, the reaction parameters were the same as those applied in the simplex reaction, except for the adjustment of the concentration of the primers and the inclusion of the primers used to amplify the control band of the 16S rDNA gene (Tables 2 and 3). The 16S L1987 and 16S H2609 (*Palumbi et al., 1991*) primers were

Freitas et al. (2023), *PeerJ*, DOI 10.7717/peerj.15364

**Table 2** Samples used in the specificity, reproducibility, and sensitivity tests of the multiplex assay of the COI gene and information of the primers used in the assay.

| Target species | *n* | Accession number | Primer | Sequence 5′–3′ | Length of the amplicon (∼bp) | Reference of the primers | Concentration (μM) |
|---|---|---|---|---|---|---|---|
| *Brachyplatystoma rousseauxii* | 14 | MT551748; OQ576109–OQ576121 | Bro385 | GGGGCCATTAACTTTATC | 254 | Present study | 0.13 |
| *Brachyplatystoma vaillantii* | 10 | MT551751–MT551753; OQ576122–OQ576128 | Bva 234 | CCTACTCCTACTCGCCTCAG | 405 | Present study | 0.07 |
| *Brachyplatystoma filamentosum* | 14 | MT551754–MT551756; OQ576098–OQ576108 | Bfi173 | CACCAGATATAGCATTCCCT | 466 | Present study | 0.07 |
| Universal (COI) | | | FishR1 | TAGACTTCTGGGTGGCCAAAGAATCA | | *Ward et al. (2005)* | 0.1 |
| Universal (rDNA 16S) | | | 16SL1987 | GCCTCGCCTGTTTACCAAAAAC | | | 0.03 |
| Universal (rDNA 16S) | | | 16SH2609 | CCGGTCTGAACTCAGATCACGT | 650 | *Palumbi et al. (1991)* | 0.03 |

**Notes.**

The pair of primers of the 16S rDNA gene was used to generate the control band in the multiplex reactions.

*n*, sample size.

Freitas et al. (2023), *PeerJ*, DOI 10.7717/peerj.15364

Peer

**Table 3** Samples used in the specificity, reproducibility, and sensitivity tests of the multiplex assay of the mitochondrial Control Region and information of the primers used in the assay.

| Target species | n | Accession number | Primer | Sequence 5′–3′ | Length of the amplicon (∼bp) | Reference of the primers | Concentration (μM) |
|---|---|---|---|---|---|---|---|
| *Brachyplatystoma rousseauxii* | 14 | MT551748; present study | Bro421 | CAGGGCCACACATTTATTT | 580 | Present study | 0.1 |
| *Brachyplatystoma vaillantii* | 10 | MT551751–MT551753; present study | Bva577 | CGCACGCTACCAATTATC | 451 | Present study | 0.13 |
| *Brachyplatystoma filamentosum* | 14 | MT551754–MT551756; present study | Bfi742 | ACCTACTATCAATCCCCCTA | 290 | Present study | 0.1 |
| Universal (Control Region) | | | F12R | GTCAGGACCATGCCTTTGTG | | *Sivasundar, Bermingham & Ortí (2001)* | 0.1 |
| Universal (rDNA 16S) | | | L2949-16S | AGTTACCCTGGGGATAAC GCGCAATC | 160 | *Kartavtsev et al. (2007)* | 0.06 |
| Universal (rDNA 16S) | | | H3058-16S | TCCGGTCTGAACTCAGATCACGTA | | | 0.06 |

**Notes.**

The pair of primers of the 16S rDNA gene was used to generate the control band in the multiplex reactions.

n, sample size.

used in the multiplex reaction of the COI (Table 2), while the L2949-16S and H3058-16S primers (*Kartavtsev et al., 2007*) were used for the CR (Table 3).

The reproducibility of the banding patterns in the two assays was tested on 10 individuals of each target species, using the same reagent concentrations, and cycling conditions as those of the multiplex reactions.

Validation tests were also applied to the multiplex protocols, using five samples of frozen fillets of each target species, which had been identified previously by *Carvalho (2018)* and *Carvalho, Sampaio & Santos (2020)*. Once again, the volumes, reagent concentrations, and cycling conditions were the same as those of the multiplex reaction.

The sensitivity of the protocols was also evaluated using different DNA concentrations of each target species (100 ng/μL, 10 ng/μL, 1 ng/μL, 0.1 ng/μL, 0.01 ng/μL). While the DNA concentrations varied, the reagent concentrations, volumes, and cycling conditions were all the same as those of the multiplex PCR.

In all the steps of the development of the multiplex assays, the banding pattern was evaluated following electrophoresis in 2% agarose gel, which was based on 3 μL of the PCR product and 3 μL of gel loading buffer containing GelRed 1x (Biotium Inc., Hayward, CA, USA). The size of these bands was determined from the migration of 7 μL of a 100-bp ladder (Sinapse Inc.; Miami, FL, USA). All the gels were electrophoresed for 80 min at 100 V, and the results were observed under an ultraviolet transilluminator (Spectroline) and photographed for analysis with a Nikon CoolPix P900.

## RESULTS

### Primer design

The species-specific forward primers for the amplification of the COI and CR sequences are described in Tables 2 and 3. These primers, combined with the FishR1 (COI) and F12R (CR) reverse primers, generated fragments of distinct sizes for each of the target taxa. Two primers were designed for the COI of *B. vaillantii*, although only the one that performed best in the specificity test was selected for the multiplex PCR assay. In the COI protocol, the selected primers amplified fragments of approximately 254 bp in the case of *B. rousseauxii*, 405 bp for *B. vaillantii*, and 466 bp for *B. filamentosum* (Table 2). Only one primer was designed for each species in the CR protocol, which generated fragments of approximately 290 bp for *B. filamentosum*, 451bp for *B. vaillantii*, and 580 bp for *B. rousseauxii* (Table 3).

### Specificity test of the primers

The results of the simplex PCR generated the banding pattern expected for the target species, and no crossover amplification was observed in the congeners or in the other catfish species commonly sold in the same markets, *i.e., Z. zungaro, P. reticulatum, P. fasciatum*, and *P. hemioliopterus* (Fig. 1). The PCR sequenced here were ≥ 99% similar to the COI, and ≥ 98% to the CR, which confirms the identification of the target species for which each primer was designed.

### Development, reproducibility, and validation of the multiplex assays

The multiplex reactions produced a double banding pattern for each of the three study species. In the case of the COI gene, a species-specific fragment of approximately 254 bp was

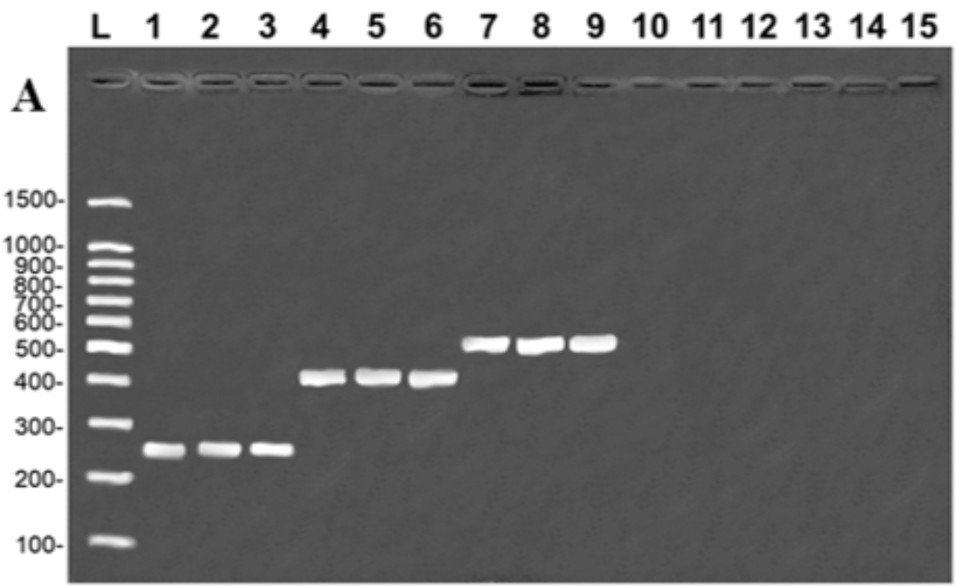

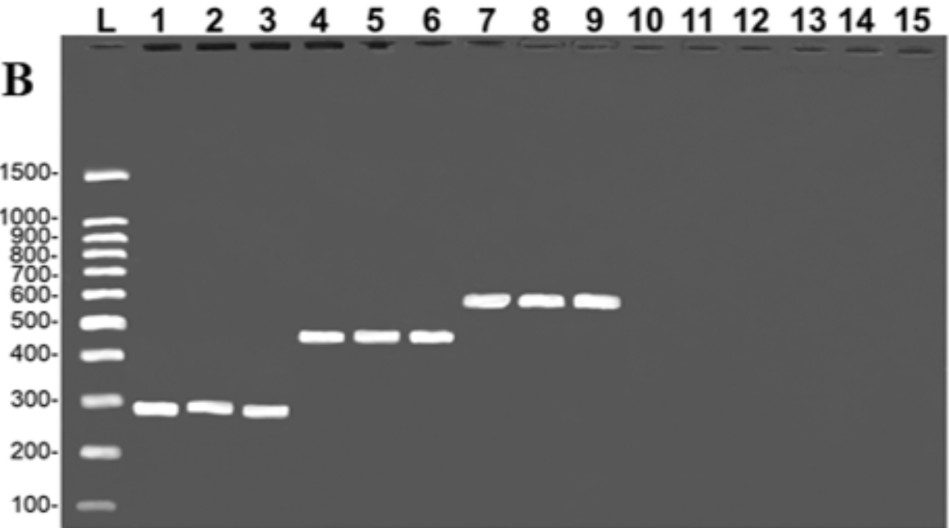

**Figure 1** **Banding pattern of the specificity tests from the COI (A) and Control Region (B) assays.** A (COI): L, 100-bp Ladder (Sinapse Inc., Richmond, Australia); 1–3, *B. rousseauxii* (254 bp); 4–6, *B. vaillantii* (405 bp); 7–9, *B. filamentosum* (466 bp). B (Control Region): L, 100-bp Ladder (Sinapse Inc., Richmond, Australia); 1–3, *B. filamentosum* (290 bp); 4–6, *B. vaillantii* (451 bp), and 7–9, *B. rousseauxii* (580 bp). In both images (A and B) lanes 10–15 represent: 10, *B. platynemum*; 11, *P. hemioliopterus*; 12, *Z. zungaro*; 13, *P. fasciatum*; 14, *P. reticulatum*; and 15, negative control.

generated for *B. rousseauxii*, 405 bp for *B. vaillantii* and 466 bp for *B. filamentosum*, with a control band of 650 bp of the 16S rDNA gene in all the reactions (Fig. 2A). The specific bands generated for the CR had approximately 290 bp in the case of *B. filamentosum*, 451 bp in *B. vaillantii*, and 580 bp in *B. rousseauxii*, with a 16S rDNA control band of approximately 160 bp (Fig. 2B). The reproducibility tests of both protocols generated

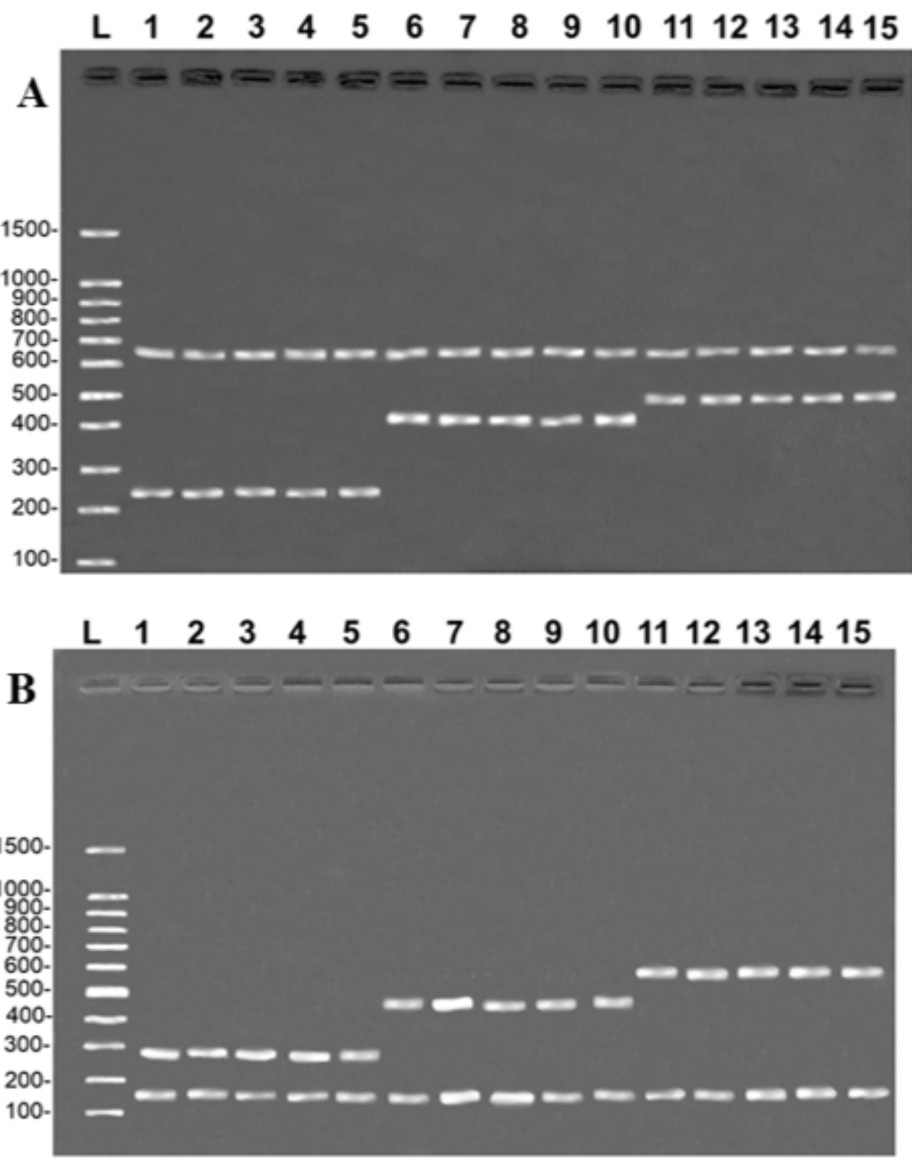

**Figure 2 Species-specific banding patterns obtained from the reproducibility tests of the multiplex protocols.** (A) (COI): L = 100-bp Ladder (Sinapse Inc., Richmond, Australia); 1–5, *B. rousseauxii* (254 bp); 6–10, *B. vaillantii* (405 bp); 11–15, *B. filamentosum* (466 bp). (B) (Control Region): L, 100-bp Ladder (Sinapse Inc., Richmond, Australia); 1–5, *B. filamentosum* (290 bp); 6–10, *B. vaillantii* (451 bp); and 11–15, *B. rousseauxii* (580 bp). The control bands of the 16S rDNA gene for the (A) COI (650 bp) and (B) Control Region (160 bp) are present in all samples.

the banding pattern expected for each of the 10 specimens tested, with the control and species-specific bands being easily identified, permitting the unequivocal identification of the target species and corroborating the efficiency and replicability of the protocol (Fig. 2).

All five samples of the fillets of each of the three target species produced the same banding patterns observed in the multiplex reactions, in both protocols, that is, the band of the species-specific fragment and the control band of the 16S rDNA (Fig. 3). This confirms

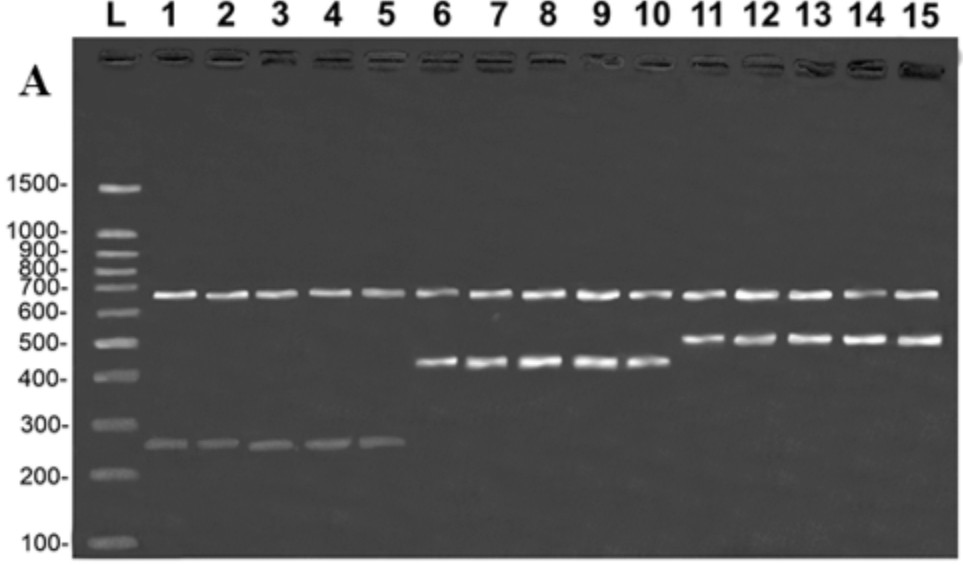

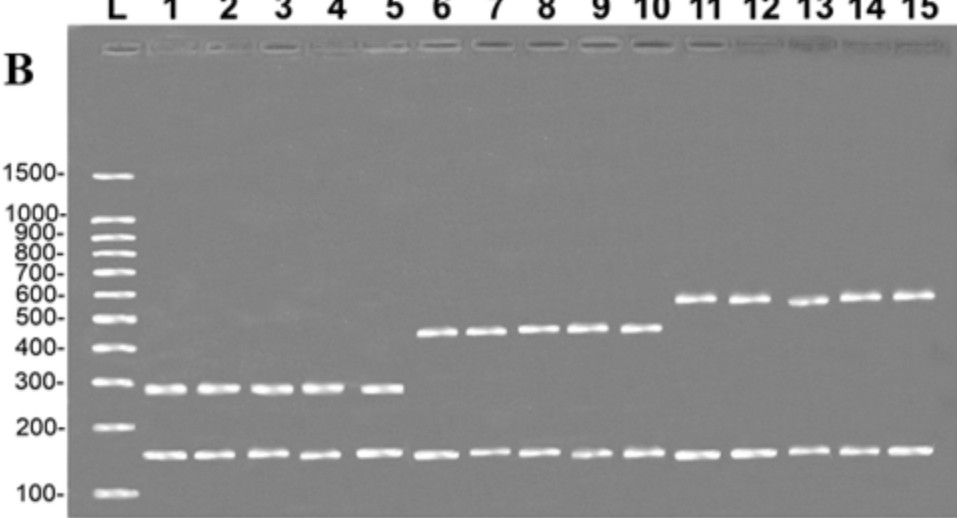

**Figure 3 Banding pattern of the validation test run on samples of fillets identified by DNA barcode.**
(A) (COI): L = 100-bp Ladder (Sinapse Inc., Richmond, Australia); 1–5, *B. rousseauxii* (254 bp); 6–10, *B. vaillantii* (405 bp); 11–15 *B. filamentosum* (466 bp). (B) (Control Region): L, 100-bp Ladder (Sinapse Inc., Richmond, Australia); 1–5, *B. filamentosum* (290 bp); 6–10, *B. vaillantii* (451 bp); and 11–15, *B. rousseauxii* (580 bp). The control bands of the 16S rDNA gene for the (A) COI (650 bp) and (B) Control Region (160 bp) are present in all samples.

that the primers developed in the present study are able to unambiguously discriminate the target species, although they have shown some degree of processing.

## Evaluation of the sensitivity of the protocols

The tests reveal that the primers developed in the present study were highly sensitive. In the case of the COI protocol, it was possible to amplify specific fragments from samples with a

DNA concentration of 0.1 ng/μL in the simple PCR (data not shown) and 1 ng/μL in the multiplex PCR (Fig. 4A). It was also possible to amplify CR fragments at a concentration of 0.1 ng/μL in the simple PCR (data not shown) and 1 ng/μL in the multiplex PCR, except for *B. vaillantii* which was only amplified from a concentration of 10 ng/μL (Fig. 4B). The fragments corresponding to the control bands of the 16S rDNA gene were amplified at concentrations of 1 ng/μL in the case of the COI gene and 0.01 ng/μL for the CR.

## DISCUSSION

A number of different DNA-based procedures have been used widely for the identification and authentication of economically important fish species (*Carvalho, 2018*; *Böhme et al., 2019*; *Carvalho, Sampaio & Santos, 2020*; *Barbosa, Sampaio & Santos, 2021*; *Wilwet et al., 2021*; *Roungchun, Tabb & Hellberg, 2022*). However, this is the first study to develop multiplex PCR protocols for the identification and authentication of the three principal catfish species of the genus *Brachyplatystoma* exploited commercially in Brazil (*B. rousseauxii, B. vaillantii,* and *B. filamentosum*), and which are the frequent targets of fraudulent substitution (*Carvalho, 2018*; *Carvalho, Sampaio & Santos, 2020*).

The differentiation of morphologically similar and phylogenetically related species, such as *B. rousseauxii, B. vaillantii,* and *B. filamentosum,* is one of the principal prerequisites for the development effective public policies, which regulate fisheries efficiently and prevent the fraudulent substitution of their products. The correct identification of fish species is not a simple task for non-experts, and often requires highly specific and sensitive techniques, especially in the case of processed fish products, which lack diagnostic traits (*Kang, 2019*; *Barbosa et al., 2020*). Previous studies have shown that the multiplex PCR is a sensitive, accurate, and efficient method for the identification of species (*Veneza et al., 2017*; *Kang, 2019*; *Barbosa et al., 2020*; *Barbosa, Sampaio & Santos, 2021*).

The selection of an appropriate molecular marker is a critical step in the development of any multiplex protocol, given that it must include regions that are highly conserved intraspecifically, while also having sufficient interspecific variability to permit the design of species-specific primers capable of discriminating the target taxa. In the present study, the COI gene and the mitochondrial Control Region both had regions of this type, which permitted the development of the primers capable of differentiating the three target species by generating species-specific banding patterns (COI: 254 bp in *B. rousseauxii*, 405 bp in *B. vaillantii*, and 466 bp in *B. filamentosum*; CR: 290 bp in *B. filamentosum*, 451 bp in *B. vaillantii*, and 580 bp in *B. rousseauxii*). The barcoding region of the COI gene is widely used for the identification of species (*Hebert, Ratnasingham & DeWaard, 2003*; *Veneza et al., 2018*; *Chang et al., 2021*) and the development of forensic protocols for the authentication of fish products (*Giusti et al., 2016*; *Veneza et al., 2017*; *Kang, 2019*; *Barbosa, Sampaio & Santos, 2021*). A number of studies have also indicated the potential of Control Region for the discrimination of taxa and have considered this region to be a promising marker for the authentication of fish products (*Sivaraman et al., 2019*; *Evangelista-Gomes et al., 2020*; *Mottola et al., 2022*; *Roungchun, Tabb & Hellberg, 2022*).

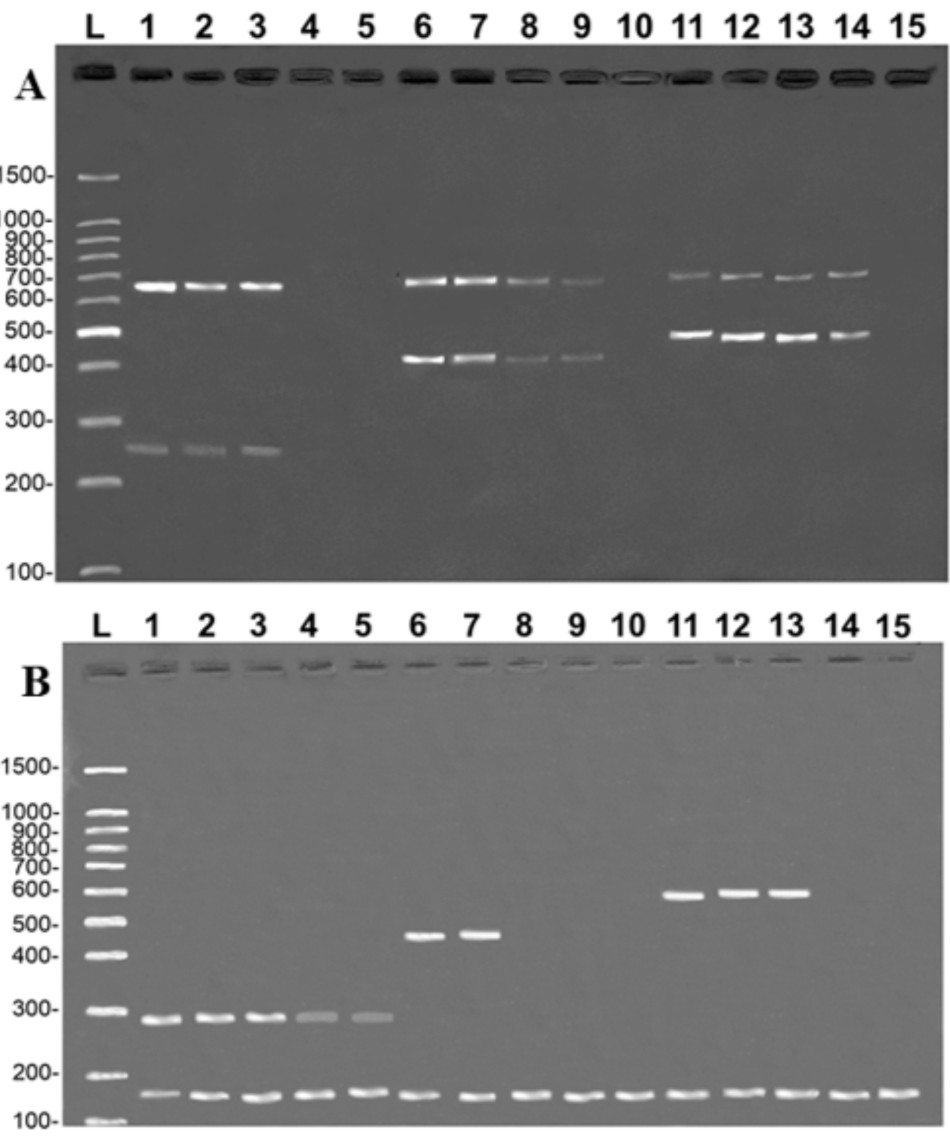

**Figure 4 Banding pattern of the sensitivity test of the multiplex assays.** The image shows the amplifications of samples based on different concentrations of DNA: 100 ng/µL (lanes 1, 6, and 11), 10 ng/µL (lanes 2, 7, and 12), 1 ng/µL (lanes 3, 8, and 13), 0.1 ng/µL (lanes 4, 9, and 14), 0.01 ng/µL (lanes 5, 10, and 15). (A) (COI): L, 100-bp Ladder (Sinapse Inc., Richmond, Australia); 1–5, *B. rousseauxii* (254 bp); 6–10, *B. vaillantii* (405 bp); 11–15, *B. filamentosum* (466 bp). (B) (Control Region): L, 100-bp Ladder (Sinapse Inc., Richmond, Australia); 1–5, *B. filamentosum* (290 bp); 6–10, *B. vaillantii* (451 bp); and 11–15, *B. rousseauxii* (580 bp). The control bands of the 16S rDNA gene for the (A) COI (650 bp) and (B) Control Region (160 bp) are present in all samples.

The protocols developed in the present study were capable of discriminating the target species through the production of two distinct electrophoretic bands, one that is species-specific and the other, a control band, which is 650 bps in the case of the COI protocol and 160 bps for the CR protocol. The use of control bands is important to ensure that the absence of PCR products is not due to the low quality or concentration of the target DNA,

malfunctions in the thermocycler or the presence of reaction inhibitors (*Barbosa, Sampaio & Santos, 2021*).

None of the primers developed in the two assays resulted in any unspecific amplification of either congeners or the other pimelodid species that are commonly sold in the same markets and are potential substitutes of *Brachyplatystoma*. An essential prerequisite for the effectiveness of a multiplex reaction is the specificity of the primer, which should not amplify the PCR products of other species, in particular, closely-related taxa, in phylogenetic terms (*Barbosa et al., 2020*; *Barbosa, Sampaio & Santos, 2021*). In this context, the specificity of the reaction is correlated with the number of differences between the species-specific primer and the non-target DNA and, in particular, the high level of complementarity of the primer with the target fragment, especially at the 3′ extremity, which permits the perfect pairing that favors the amplification of the species-specific fragment for the discrimination of the taxa (*Ali, Razzak & Hamid, 2014*; *Kang, 2019*; *Li et al., 2019*; *Barbosa, Sampaio & Santos, 2021*). In the assays developed in the present study, the primers presented differences between the congeners and other pimelodids that varied from 5% to 30% when using the COI gene, and 10% to 40% in the case of the Control Region, with most of the differences being concentrated in the 3′ end. The stoichiometric parameters, that is, the concentration and volume of the reagents, and the cycling conditions were optimized to ensure the specificity of the multiplex protocols. Previous studies have demonstrated that the accentuated complementarity of the primers, the stoichiometry of the reaction, and the cycling conditions are all essential to ensure the successful discrimination of species in multiplex assays (*Ali, Razzak & Hamid, 2014*; *Barbosa et al., 2020*; *Barbosa, Sampaio & Santos, 2021*).

The results of the present study also demonstrated that the protocols were sensitive enough for the amplification of the fragments of the target species at DNA concentrations as low as 1 ng/μL, with the exception of the CR of *B. vaillantii*, which was only detected at a concentration of 10 ng/μL. On the other hand, the control band of the 16S rDNA gene amplified fragments at concentrations of as low as 0.01 ng/μL for the Control Region, and 1 ng/μL for the COI. Frozen and highly processed foods are more susceptible to the fragmentation and low concentrations of DNA, which can affect their amplification by PCR (*Böhme et al., 2019*; *Adibah et al., 2020*). The sensitivity tests were run on the DNA of frozen fish, which may account for the differences in the detection limits. In general, the multiplex protocols are able to differentiate the species even when the DNA is present in low concentrations, which indicates that they are sensitive methods for the authentication of fish, whether sold either fresh or frozen.

*Carvalho (2018)* and *Carvalho, Sampaio & Santos (2020)* used DNA barcoding to authenticate processed fillets of *B. rousseauxii*, *B. vaillantii*, and *B. filamentosum*, and while this approach was efficient, it is based on the full sequencing of the target region, which requires sophisticated equipment, special training, and specific reagents that all increase the time and costs of the identification process. By contrast, multiplex PCR has fewer steps, and requires less equipment and reagents, given that it requires only the extraction of the DNA, the amplification of the target fragment, and gel electrophoresis. Overall, then, the protocols developed in the present study are faster and more cost-effective

than authentication by DNA barcoding. The reproducibility tests and the validation of these multiplex assays also indicated that this approach is accurate and efficient for the unequivocal discrimination of the target species, whether sold fresh or in processed form.

Methods used to authenticate foods must be specific, reliable, precise, and have a multiplex capacity, in order to identify a number of different samples in a single analysis (*Böhme et al., 2019*). In this context, recently-developed technologies, such as NGS, DNA-metabarcoding, BAR-HRM, loop-mediated isothermal amplification (LAMP), and ddPCR, have proved promising, and may substitute the more traditional methods, given that they permit rapid, multiple identifications (*Fernandes et al., 2018*; *Böhme et al., 2019*; *Noh et al., 2021*; *Xu et al., 2022*). Despite their advantages, including their high yield and sensitivity, these modern methods require more complex laboratories, qualified personnel, and expensive reagents (*Creydt & Fischer, 2020*; *Pappalardo et al., 2022*). Multiplex PCR, by contrast, is one of the traditional methods most used in forensic investigations, given its sensitivity, specificity, rapid throughput, relatively simple procedures, and low laboratory costs (*Veneza et al., 2017*; *Barbosa, Sampaio & Santos, 2021*; *Wilwet et al., 2021*). Given this, the multiplex PCR protocols developed in the present study can be used routinely by fish processing plants to certify their products, and by government inspectors, to prevent fraudulent substitutions and mislabeling in the productive chain.

## CONCLUSIONS

The present study is the first to develop multiplex PCR assays for the reliable discrimination of three species of catfish (*B. rousseauxii*, *B. vaillantii*, and *B. filamentosum*) that are economically important in Brazil and are often the target of fraudulent commercial substitutions. The two protocols developed were accurate, efficient, and sensitive for the unequivocal identification of the target species and can be used to authenticate the species from either fresh or processed samples. These multiplex PCR protocols are also rapid and cost-effective, and can be used by fish processing plants to certify their products, and by government agencies to prevent fraud in the retailing of fish and fish products.

### Funding
The present study was funded by the Brazilian National Council for Scientific and Technological Development, CNPq (grant 407536/2021-3), and in part by the Brazilian Coordination for Higher Education Personnel Training (CAPES)—Finance Code 001, through the concession of a graduate (masters) scholarship to Leilane de Freitas Brito. The PROPESP-UFPA (PAPQ) funded the article publication fee. The funders had no role in study design, data collection and analysis, decision to publish, or preparation of the manuscript.

### Grant Disclosures
The following grant information was disclosed by the authors:

Brazilian National Council for Scientific and Technological Development, CNPq: 407536/2021-3.

Brazilian Coordination for Higher Education Personnel Training (CAPES)—Finance Code 001, through the concession of a graduate (masters) scholarship to Leilane de Freitas Brito. The PROPESP-UFPA (PAPQ) funded the article publication fee.

## Competing Interests

The authors declare there are no competing interests.

## Author Contributions

- Leilane Freitas conceived and designed the experiments, performed the experiments, analyzed the data, prepared figures and/or tables, authored or reviewed drafts of the article, and approved the final draft.
- Andressa J. Barbosa performed the experiments, analyzed the data, prepared figures and/or tables, authored or reviewed drafts of the article, and approved the final draft.
- Bianca A. Vale performed the experiments, analyzed the data, prepared figures and/or tables, authored or reviewed drafts of the article, and approved the final draft.
- Iracilda Sampaio analyzed the data, authored or reviewed drafts of the article, and approved the final draft.
- Simoni Santos conceived and designed the experiments, performed the experiments, analyzed the data, prepared figures and/or tables, authored or reviewed drafts of the article, and approved the final draft.

## Animal Ethics

The following information was supplied relating to ethical approvals (*i.e.,* approving body and any reference numbers):

We collect samples from fish markets, these products were already intended for human consumption. Therefore, as we do not collect live animals, in nature or from fish farms, we do not submit the research for ethical approval by the Ethics Committee of the Federal University of Pará.

## Data Availability

The sequences are available at GenBank (Table 1):

- *Brachyplatystoma rousseauxii*: MT551748; FJ418759; OQ576109 –OQ576121
- *Brachyplatystoma vaillantii*: MT551751–MT551753; KT952409; OQ576122–OQ576128
- *Brachyplatystoma filamentosum*: MT551754–MT551756; OQ576098–OQ576108
- *Brachyplatystoma tigrinum*: KT952408
- *Brachyplatystoma juruense*: KT952405; KR491564
- *Brachyplatystoma platynemum*: KT952406; OQ576135; OQ576136
- *Brachyplatystoma capapretum*: KT952403
- *Pseudoplatystoma reticulatum*: GU570868; GU570869; OQ576137; OQ576138
- *Pseudoplatystoma magdaleniatum*: GU570859; GU570860

- *Pseudoplatystoma fasciatum*: GU570849; OQ576139
- *Pseudoplatystoma corruscans*: HQ600841; HQ600842; JX462914
- *Pseudoplatystoma metaense*: JQ733555
- *Pseudoplatystoma orinocoense*: JQ733556
- *Pseudoplatystoma tigrinum*: GU570936; GU570937
- *Pseudoplatystoma punctifer*: KT952427
- *Sorubimichthys planiceps*: GU570940; GU570941
- *Hemisorubim platyrhynchos*: GU570707; KT952411
- *Zungaro zungaro*: KT952431; KP294234; OQ576129–OQ576131
- *Zungaro jahu*: EU179810; JN813033
- *Phractocephalus hemioliopterus*: KP772589; OQ576132–OQ576134.

## Supplemental Information

Supplemental information for this article can be found online at http://dx.doi.org/10.7717/peerj.15364#supplemental-information.

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
