# Peer review of "Development of rapid and cost-effective multiplex PCR assays to differentiate catfish of the genus Brachyplatystoma (Pimelodidae–Siluriformes) sold in Brazil"

_PeerJ, doi:10.7717/peerj.15364_

## Round 0.1 · original submission · Major Revisions

The authors are encouraged to kindly attend to all the comments raised by reviewers. Importantly, authors should provide as much detail as possible, not only in the revised manuscript but also in the responses to reviewers' comments.

I look forward to your revised manuscript. Thank you

Reviewer 1 ·

Basic reporting

This is an exciting first study to develop multiplex PCR protocols for the identification and authentication of the three principal catfish species to prevent fraudulent commercial substitutions in Brazil. The authors have collected a unique dataset using Multiplex PCR assays. Overall the paper is well-written and structured except for lines 34-35: Kindly rewrite the sentence.

Experimental design

No comments

Validity of the findings

The methodology was painstakingly detailed and incorporated a sufficient number of samples. There were clear scientific and practicality to support the recommendation of the multiplex PCR assay to determine the target species both at the fresh and frozen levels.

Additional comments

This is an excellent report, and I approve of its publication. It is comprehensive and
completely justifies the author's recommendation for the Development of rapid and cost-effective multiplex 1 PCR assays to differentiate catfish genus sold in Brazil.

Reviewer 2 ·

Basic reporting

Clear and unambiguous, professional English used throughout.

Literature references, sufficient field background/context provided.

Professional article structure, figures, tables. Raw data shared, however, the DNA sequences must be submitted to public databases, so that the codes are inserted in the article, before its approval.

Self-contained with relevant results to hypotheses.

Experimental design

Original primary research within Aims and Scope of the journal.

Research question well defined, relevant & meaningful. It is stated how research fills an identified knowledge gap.

Rigorous investigation performed to a high technical & ethical standard.

Methods described with sufficient detail & information to replicate.

However, the small number of individuals from other species marketed together with the target species and/or with close phylogenetic relationships, used for specificity tests and primer design, may reduce the effectiveness of the developed primers, considering the intraspecific variation of the fragments used, as detailed below.

Validity of the findings

No comment.

Additional comments

The article refers to the development of two identification protocols for important freshwater catfish species commercialized in Brazil, based on multiplex PCR. The work is original and relevant, considering that the method is efficient and cost-effective, applied to species for which there are already records of commercial fraud.

However, small improvements can be implemented in order to better justify the sampling.
For example, between lines 122 to 124, the numbers of samples used per species, for the specificity tests, are small, considering the possibility of intrapopulation variation, mainly, but not only, for the Control Region.

In addition, in Table 1 I verified that for the design of the primers, in some cases, a reduced number of sequences were also used (for example, n=1 for some species of Brachyplatystoma). The authors could justify that although the number of sequences is reduced, they refer to different haplotypes (if applicable), that is, that they used a larger number of sequences and were careful to select sequences from different haplotypes, as a way to to expand the possibility of capturing possible intraspecific variations and thus obtaining more specific primers.

In line 159, the authors describe that 36 COI sequences from other pimelodid were used and included in the database. In line 125, the authors had already mentioned 35 sequences and in Table 1, there were also 35 sequences. Correct this.

On lines 160-161, the authors describe that for the design of the specific primers for the control region, at least three sequences of the target species were used. Considering that the control region is a region of recognized intraspecific variability, wouldn't three sequences for each target species be a very small number? Do these sequences represent different haplotypes, obtained from a larger number of sequences? Explain better about it.

In addition to these adjustments recommended above, for improvements in the description of the methodology, I believe that small text adjustments can also be made to the results. On line 259, the authors describe the efficiency of the primers developed in the study, including for processed products. However, they use frozen fillets in the validation of the protocols, which represent a minimally processed product, so that there was no heat treatment or use of additives such as those used in dishes prepared in restaurants. Thus, the sentence in line 259 must be rewritten so that it does not appear that the protocol can be successfully used for any processed product, considering that there are different degrees of processing and this can influence the efficiency of the protocol. Writing suggestion: "This confirms that the primers developed in the present study are able to unambiguously discriminate the target species, although they have been presented with some degree of processing."

And finally, in the caption of Figure 4, in the third line, the DNA concentration in the sensitivity test must be corrected from 0.001 ng/μL to 0.01 ng/μL, as described in line 212, when the authors explain the experimental design for the performance of specificity tests.

Reviewer 3 ·

Basic reporting

1.1. For the purpose of clarity, several terms should be disambiguated when they are first introduced. Ribosomal DNA as rDNA, the Cytochrome C oxidase I gene as COI, etc.

1.2. While this may be common knowledge in the subfields of DNA-based identification and forensics, the manuscript would be more accessible if the introduction section included an explanation on the rationale behind using mitochondrial DNA rather than genomic DNA (e.g., abundance, robustness of assay, benefit of building on common practices in the field, etc.)

1.3. Likewise, it would also be beneficial to provide some details about the rationale behind choosing the COI and CR regions for this assay.

Experimental design

No comment

Validity of the findings

3.1. Please include raw data from Sanger sequencing experiments as supplementary information. This would improve replicability and lend further credibility to the species identification process.

Reviewer 4 ·

Basic reporting

- Clear and unambiguous, professional English
- Literature references are complete

Experimental design

These multiple PCR methods should also be tested on other species of fish besides pimelodid taxa.

Validity of the findings

No comment

Additional comments

This research is very valuable for identification three fish species in the genus Brachyplatystoma using simple, rapid, and economical method in both fresh and processed samples. However, there are some comments that need to be revised as follows.
1. Lines 28 and 30: Please define the abbreviation completely at first mention. COI (Cytochrome c oxidase I) and CR (Control Region).
2. Line 125: Change “35 COI” to “36 COI”
3. How many samples of each species were investigated in the simplex PCR, 5 or 3? I'm confused because Line 178 used five specimens of each target species, but Line 195, three positive samples of each marker from each target species…. Also, Figure 1 showed banding pattern from 3 samples of each species.
4. Lines 200-203: Why did not use the same primers as the control band of the 16S rDNA in both COI and CR?
5. Lines 204-205 and 252: Why did you specify 10 individuals of each target species, but in Figure 2 there were 5 samples of each species?
6. Lines 206, 210 and 214: Change “multiplex” to “simplex”. Because the reaction and thermal cycling of multiplex PCR did not mention but carried on following the simplex PCR.
7. Line 216: …3 µL of the PCR and… revise to …3 µL of the PCR product and…
8. Lines 239-240: Please revise to “…i.e., B. platynemum, P. hemioliopterus, Z. zungaro, P. fasciatum, and P. reticulatum.”
9. Lines 295-296, 307, the figure titles of Figure 1-4: The difference sizes of PCR products must be specified, did not use “~”.
10. Lines 323-324: How did the authors calculate (5-30% and 10-40%)?
11. Figure 4: …0.001 ng/μL (lanes 5, 10, and 15). Change to 0.01
12. Table 1: (1) n (COI) of Pseudoplatystoma reticulatum should be 4, not 5. Please check.
: (2) Please check n (CR) of Brachyplatystoma rousseauxii, Brachyplatystoma vaillantii, and Brachyplatystoma filamentosum, 6 or 3?
13. Please revise L2949-16S and H3058-16S to 16S L2949 and 16S H3058.
14. Please check how to refer to equipment or reagents (manufacturer’ s names, their city, and country).

---

## Round 0.2 · Minor Revisions

Please authors, kindly attend to the issues raised by the reviewers, paying close attention to providing details in the revised manuscript.

Reviewer 2 ·

Basic reporting

No comment.

Experimental design

No comment.

Validity of the findings

No comment.

Additional comments

The authors responded to most of the requested corrections and, when this was not possible, satisfactorily justified. However, they should observe the legend of Figure 4, on the third line, in which the DNA concentration in the sensitivity test must be corrected from 0.001 ng/μL to 0.01 ng/μL, as had already been requested in the first correction.

Reviewer 3 ·

Basic reporting

The authors satisfactorily addressed all comments.

Experimental design

The authors satisfactorily addressed all comments.

Validity of the findings

The authors satisfactorily addressed all comments.

Reviewer 4 ·

Basic reporting

This manuscript has been revised by the authors as suggested by the reviewer. However, there are still 2 comments that are not yet. Please consider making corrections as follows:

1. Lines 307-308 and 319: The difference sizes of PCR products must be specified, did not use “~”.

2. Figure 4: …0.001 ng/μL (lanes 5, 10, and 15). Change to 0.01

Experimental design

-

Validity of the findings

-

Additional comments

After revision the 2 comments, this manuscript can be accept.

---

## Round 0.3 · accepted · Accept

Thank you authors for thoroughly addressing all concerns of the reviewers.
I am very satisfied with the revised manuscript. It is acceptable for publication.

Thank you authors for finding PeerJ as your journal of choice, and I look forward to your future scholarly contributions.